# Analysis of the Function of the Lymphocytic Choriomeningitis Virus S Segment Untranslated Region on Growth Capacity In Vitro and on Virulence In Vivo

**DOI:** 10.3390/v12080896

**Published:** 2020-08-16

**Authors:** Satoshi Taniguchi, Tomoki Yoshikawa, Masayuki Shimojima, Shuetsu Fukushi, Takeshi Kurosu, Hideki Tani, Aiko Fukuma, Fumihiro Kato, Eri Nakayama, Takahiro Maeki, Shigeru Tajima, Chang-Kweng Lim, Hideki Ebihara, Shigeru Kyuwa, Shigeru Morikawa, Masayuki Saijo

**Affiliations:** 1National Institute of Infectious Diseases, 1-23-1 Toyama, Shinjuku-ku, Tokyo 162-8640, Japan; rei-tani@nih.go.jp (S.T.); ytomoki@nih.go.jp (T.Y.); shimoji-@nih.go.jp (M.S.); fukushi@nih.go.jp (S.F.); kurosu@nih.go.jp (T.K.); toyamaeiken3@juno.ocn.ne.jp (H.T.); fukuma-aiko@pref.shimane.lg.jp (A.F.); fumihiro@nih.go.jp (F.K.); nakayama@nih.go.jp (E.N.); tomaeki@nih.go.jp (T.M.); stajima@nih.go.jp (S.T.); ck@nih.go.jp (C.-K.L.); 2Toyama Institute of Health, 17-1 Nakataikoyama, Imizu-shi, Toyama 939-0363, Japan; 3QIMR Berghofer Medical Research Institute, 300 Herston Rd, Herston, Queensland 4006, Australia; 4Mayo Clinic, 200 First St. SW, Rochester, MN 55905, USA; Ebihara.Hideki@mayo.edu; 5Graduate School of Agricultural and Life Sciences, The University of Tokyo, 1-1-1 Yayoi, Bunkyo-ku, Tokyo 113-8657, Japan; akyuwa1@g.ecc.u-tokyo.ac.jp; 6Okayama University of Science, 1-3 Ikoinooka, Imabari-shi, Ehime 794-8555, Japan; s-morikawa@vet.ous.ac.jp

**Keywords:** Lymphocytic choriomeningitis virus, LCMV strain WE, arenavirus, reverse genetics, untranslated region

## Abstract

Lymphocytic choriomeningitis virus (LCMV) is a prototypic arenavirus. The function of untranslated regions (UTRs) of the LCMV genome has not been well studied except for the extreme 19 nucleotide residues of both the 5′ and 3′ termini. There are internal UTRs composed of 58 and 41 nucleotide residues in the 5′ and 3′ UTRs, respectively, in the LCMV S segment. Their functional roles have yet to be elucidated. In this study, reverse genetics and minigenome systems were established for LCMV strain WE and the function of these regions were analyzed. It was revealed that nucleotides 20–40 and 20–38 located downstream of the 19 nucleotides in the 5′ and 3′ termini, respectively, were involved in viral genome replication and transcription. Furthermore, it was revealed that the other internal UTRs (nucleotides 41–77 and 39–60 in the 5′ and 3′ termini, respectively) in the S segment were involved in virulence in vivo, even though these regions did not affect viral growth capacity in Vero cells. The introduction of LCMV with mutations in these regions attenuates the virus and may enable the production of LCMV vaccine candidates.

## 1. Introduction

Lymphocytic choriomeningitis virus (LCMV) belongs to the genus *Mammarenavirus*, family *Arenaviridae*. There are two groups of *Mammarenavirus*, new-world and old-world arenaviruses [1]. LCMV belongs to the old-world arenavirus group, as does Lassa virus (LASV), the causative agent of Lassa fever. LCMV infects humans, causing flu-like fever, nausea, neck stiffness, headache, and occasionally photophobia; meningitis and encephalitis appear in severe cases [2,3,4,5,6]. Humans may be infected with LCMV if they are exposed to the body fluids of infected house mice (*Mus musculus*), which is the natural reservoir of LCMV.

LCMV was first isolated in 1933, and thus, it has been the most widely used model system for the study of viral immunology, persistent infection, and pathogenesis [7,8,9]. The LCMV strain Armstrong (LCMV-ARM), a laboratory strain that causes acute neurotropic infection in mice, has been commonly used in these studies. Minigenome and reverse genetics systems based on LCMV-ARM are powerful tools for the study of LCMV-ARM pathogenesis and propagation mechanisms [10,11,12,13,14,15,16,17]. However, molecular biological analyses based on other strains of LCMV have not yet been performed.

Infection with different strains of LCMV causes different manifestations of disease in rodent models [18,19,20,21]. For example, the LCMV-ARM causes neurotropic symptoms in mice, while the LCMV strain WE (LCMV-WE) causes viscerotropic symptoms in mice. Additionally, different manifestations of disease caused by infection with different strains of LCMV are observed in the nonhuman primates (NHPs) model. Rhesus macaques infected with LCMV-ARM are asymptomatic, while those with LCMV-WE have symptoms related to hemorrhagic fever, hepatic damage, and meningitis [22,23,24,25,26,27]. The infection of NHPs with LCMV-WE is considered a suitable animal model for provision of insights into Lassa fever in humans [26]. However, despite the unique and important nature of this virus, basic tools that are necessary to analyze the virologic characteristics of LCMV-WE, such as a reverse genetics system of LCMV-WE, have not yet been developed.

The genome of LCMV consists of two negative-sense single-stranded RNA segments, designated S and L. The S segment is approximately 3.4 kilobases (kb) in length and encodes a viral glycoprotein precursor (GPC) and a nucleoprotein (NP), while the L segment, of approximately 7.2 kb, encodes a viral RNA-dependent RNA polymerase (L) and a polypeptide that contains a small zinc finger-domain (Z) [1]. Each segment utilizes an ambisense coding strategy in which the two open reading frames are encoded in opposite orientations, terminating in a central intergenic region (IGR) that may fold into a predictable and stable secondary structure. It was reported that replacement of the L segment IGR with the S segment IGR made the recombinant virus highly attenuated and induced protective immunity against a lethal wildtype LCMV (wtLCMV) challenge [28]. The results of this report suggested that noncoding regions of LCMV gene were involved in virulence; as such, these noncoding regions may serve as targets to promote viral attenuation for vaccine development.

Untranslated regions (UTRs) are located at the termini of each genome segment, and the 19 nucleotides (nt) of the extreme 5′ and 3′ termini of both S and L segments are essential for viral transcription and replication [29,30]. These regions comprise complementary base pairs and are recognized as promoters of L-polymerase-driven viral genome transcription and replication [29,30]. However, the functional roles in virulence and propagation of other regions within the UTRs have yet to be elucidated. To investigate the roles of these regions, we developed reverse genetics and minigenome systems for LCMV-WE. The secondary structures of the RNA sequences of the 5′- and 3′-terminal UTRs of the S segment were predicted. Several variants of infectious clones containing S segment UTRs lacking their internal regions were generated, and the viral propagation efficiency of these clones was evaluated. The virulence of these infectious clone variants in mice was also assessed. Furthermore, we generated variants of minigenome RNAs including cognate deletions and elucidated their efficiency in viral genome replication, transcription, and packaging of virus-like particles (VLPs).

## 2. Materials and Methods

### 2.1. Cells and Viruses

A549, baby hamster kidney fibroblast (BHK-21), and Vero cells were maintained in Dulbecco’s modified Eagle medium (DMEM) supplemented with 5% fetal bovine serum (FBS) and 100 µg/mL penicillin–streptomycin (all from Life Technologies, Carlsbad, CA, USA) (DMEM-5FBS) and cultured at 37 °C in a 5% CO_2_ atmosphere. LCMV strain WE-NIID (GenBank accession numbers LC413283 and LC413284) was amplified in Vero cells and used in this study.

### 2.2. Immunofocus Assay and Immunofluorescence Assay (IFA)

The infectious dose of LCMV was determined using a viral immunofocus assay. Briefly, after absorption of virus solution into Vero cells cultured in 12-well plates, cells were further cultured for 120 h at 37 °C in DMEM supplemented with 1% FBS and 100 µg/mL penicillin–streptomycin (DMEM-1FBS) with agarose (1%). The cell monolayers were then fixed with 10% formalin in phosphate buffered saline (PBS), permeabilized by incubating with 0.2% Triton X-100 in PBS, and stained with anti-LCMV-WE recombinant NP immunized rabbit serum and horseradish peroxidase (HRP)-goat anti-rabbit IgG (H+L) DS Grd (lot: 917439A, Life Technologies) [31]. Cells were then stained with Peroxidase Stain DAB Kit (Nacalai, Kyoto, Japan), and the number of stained foci was counted. For immunofluorescence assay (IFA), Alexa Fluor 488 goat anti-rabbit IgG (H+L) (Life Technologies) was used as the secondary antibody. The cells were observed to determine if they were LCMV-positive or -negative under a fluorescence microscope (BZ-9000, Keyence, Osaka, Japan).

### 2.3. Plasmids

To construct pRF-WE-SRG and pRF-WE-LRG plasmids, cDNA fragments containing either whole S or L segments were cloned between the murine pol I promoter and the terminator of the pRF vector. The pRF vector system was kindly provided by Shuzo Urata, Nagasaki University (Nagasaki, Japan) and Juan Carlos de la Torre, of the Scripps Research Institute (San Diego, CA, USA) [17]. Insertion of additional G residue directly downstream of the promoter has been reported to enhance the efficiency of both reverse genetics and minigenome systems. The viral cDNA constructs were inserted in the sense orientation for viral complementary (c)RNA. The following pRF-WE-SRG plasmids with mutations in the S segment UTRs were generated via site-directed mutagenesis: pRF-WE-SRG-5UTRΔ20–40; pRF-WE-SRG-5UTRΔ41–60; pRF-WE-SRG-5UTRΔ60–77; pRF-WE-SRG-3UTRΔ20–38; pRF-WE-SRG-3UTRΔ39–60; pRF-WE-SRG-UTR-comple; pRF-WE-SRG-UTR reverse; pRF-WE-SRG-UTR 5-3 change; pRF-WE-SRG-Δ26–40; pRF-WE-SRG-Δ20–25; pRF-WE-SRG-Δ20–30; and pRF-WE-SRG-Δ31–40.

To construct SMG-luc and SMG-green fluorescent protein (SMG-GFP) as minigenome plasmids (SMGs), cDNA fragments containing the S 5′ UTR, the S IGR, and either Renilla luciferase (luc) or GFP open reading frames (ORFs) in the antisense orientation to the 5′ UTR and the S 3′ UTR were cloned between the murine pol I promoter and the terminator of the pRF vector in sense orientation to the viral cRNA. An additional G residue was also inserted between the murine pol I promoter and the viral minigenome sequence as well as the reverse genetics system. SMG-luc or -GFP plasmids with mutations in their S segment UTRs were generated by site-directed mutagenesis: SMG-5UTRΔ20–40 (-luc or -GFP); SMG-5UTRΔ41–60 (-luc or -GFP); SMG-5UTRΔ60–77 (-luc or -GFP); SMG-3UTRΔ20–38 (-luc or -GFP); SMG-3UTRΔ39–60 (-luc or -GFP); and SMG-UTR-comple (-luc).

Detailed information for each mutated pRF-WE-SRG and SMG-luc or -GFP is shown in Table 1 and Table 2. Briefly, pRF-WE-SRG (or SMG)-5UTRΔ20–40, pRF-WE-SRG (or SMG)-5UTRΔ41–60, and pRF-WE-SRG (or SMG)-5UTRΔ60–77 lacked nt between 20–40, 41–60, and 60–77 in the S segment UTRs of their 5′ terminus, respectively. pRF-WE-SRG (or SMG)-3UTRΔ20–38 and pRF-WE-SRG (or SMG)-3UTRΔ39–60 lacked nt between 20–38 and 39–60 in the S segment UTRs of their 3′ terminus, respectively. pRF-WE-SRG (or SMG)-UTR-comple contained complementary nucleic sequences in nt 20–26 of their 5′ terminus and 20–24 of their 3′ terminus, respectively. pRF-WE-SRG-UTR 5-3 change contained nt 20–38 in the 3′ terminus in place of nt 20–40 in the 5′ terminus and vice versa in the 3′ terminus. pRF-WE-SRG-Δ26–40, pRF-WE-SRG-Δ20–25, pRF-WE-SRG-Δ20–30, and pRF-WE-SRG-Δ31–40 lacked nt 26–40, 20–25, 20–30, and 31–40 in the S segment UTRs of their 5′ terminus, respectively; these plasmids also lacked nt 24–38, 20–23, 20–28, and 29–38 in the S segment UTRs of their 3′ terminus, respectively.

Plasmids for the expression of LCMV-WE NP, L, Z, and GPC (referred to as pC-NP, pC-L, pC-Z, and pC-GPC, respectively) were constructed by cloning PCR amplicons of the NP, Z, and GPC genes flanked by EcoRI and NheI sites and of the L gene flanked by KpnI and NheI sites into their respective restriction enzyme sites in the pCAGGS plasmid.

A plasmid expressing GFP (referred to as pC-GFP) was constructed by cloning PCR amplicons encoding GFP genes flanked by EcoRI restriction sites into a compatible site in pCAGGS plasmid. A plasmid expressing firefly luciferase (referred to as pC-Fluc) was kindly provided by Hideki Aizaki of the National Institute of Infectious Diseases (Tokyo, Japan).

### 2.4. Transfection, Minigenome, and Reverse Genetics System

For the minigenome assay, 1.0 × 10^5^ BHK-21 cells were seeded into wells of 24-well plates and grown to approximately 80% confluence. Cells were transfected with minigenome plasmids (0.24 µg of SMG-luc or SMG-GFP), 0.24 µg of pC-NP or pCAGGS vector (empty vector), 0.3 µg of pC-L or empty vector, and 0.03 µg of pC-Fluc; transfections were conducted using TansIT-LT1 DNA transfection reagent (Mirus Bio, Madison, WI, USA). The total amount of DNA used in transfections was maintained at 0.81 µg, and the amount of transfection reagent was maintained at a constant ratio of three volumes (µL) per amount of DNA added (µg). Transfected cells were incubated for two days at 37 °C. GFP expression was detected by fluorescence microscope; and renilla and firefly luciferase activities were measured using a Renilla Luciferase Assay System (Promega, Fitchburg, WI, USA) and a Bright-Glo Luciferase Assay System (Promega), respectively. The detailed methodology is described in the luciferase assay section.

For the reverse genetics system, 3.0 × 10^5^ BHK-21 cells were seeded into 6-well plates to reach approximately 80% confluence. Cells were transfected using TansIT-LT1 DNA transfection reagent with 0.8 µg pRF-WE-SRG, 1.4 µg pRF-WE-LRG, 0.8 µg pC-NP, and 1.0 µg pC-L. The total amount of DNA transfected was 4.0 µg, and the amount of transfection reagent used was 12.0 µL. Under these transfection conditions, the cells were incubated for at least 3 days at 37 °C. The supernatant was harvested, and the infectious dose of recombinant virus was measured using an immunofocus assay. The RNA of each recombinant LCMV (rLCMV) was extracted, and viral genome sequences of all rLCMVs were confirmed to be as intended using the Sanger Sequence method.

### 2.5. Rescue of LCMV RNA Analogs into LCMV-Like Particles

The rescue of LCMV RNA analogs into VLPs was carried out as previously reported [11], with 3.0 × 10^5^ BHK-21 cells in 2.0 mL of DMEM-5FBS seeded into 6-well plates to reach approximately 80% confluence. The cell culture medium was changed to DMEM with 2% FBS and 100 µg/mL penicillin–streptomycin (DMEM-2FBS). The cells were transfected using TansIT-LT1 DNA transfection reagent with 1.0 µg SMGs-luc or -GFP (SMG-luc or -GFP or various mutated SMGs-luc or -GFP), 0.8 µg pC-NP, 1.0 µg pC-L, 0.1 µg pC-Z, and 0.3 µg pC-GPC. As a background control for the rescue of LCMV RNA analogs into VLPs, cells transfected with 1.0 µg SMG, 0.8 µg pC-NP, and 1.0 µg pC-L, which were not expected to produce VLPs without the VLP expression plasmids (pC-Z and pC-GPC), were used (SMG-luc, (no VLPs); SMG-GFP, (no VLPs)). Cells transfected with 1.0 µg pC-GFP were used to control for transfection efficiency. The amount of transfection reagent was kept to three volumes (µL) per amount of DNA added (µg). After incubation for 48 h at 37 °C with 5% CO_2_, the expression of renilla luciferase or GFP was examined and then 1.2 mL of supernatant from each well was harvested. Fresh monolayers of BHK-21 cells seeded into 6-well plates were infected with the supernatants. Cells were incubated with the supernatants for 4 h at 37 °C and infected with helper LCMV at a multiplicity of infection (MOI) of 2 focus-forming units (FFU)/cell. Ninety hours postinfection, the cells were examined for renilla luciferase or GFP expression.

### 2.6. RNA Secondary Structure Prediction

To predict RNA secondary structures, the web application CENTROIDFOLD was used (https://www.ncrna.org) [32]. The RNA sequences of the 5′ and 3′ termini UTRs and 50 nt lengths of ORF (GPC or NP ORF) regional RNA sequences, which were directly downstream of the UTRs, were linked and sent to the CENTROIDFOLD server. The CONTRAfold model (weight of base pairs: 2^2^) was used to calculate base-pairing probabilities. The results of RNA secondary structure prediction were referenced in the generation of rLCMVs, in which mutations were introduced into the UTRs.

### 2.7. Luciferase Assay

Cells were washed with PBS and lysed with 100 µL (24-well plates) or 500 µL (six-well plates) Renilla Luciferase Assay System Lysis Buffer (Promega). To measure renilla luciferase activity, cell lysates (20 µL) were mixed with 100 µL Renilla Luciferase Assay System Substrate (Promega). To measure firefly luciferase activity, cell lysates (2 µL) were mixed with 18 µL Renilla Luciferase Assay System Lysis Buffer and 50 µL Bright-Glo Luciferase Assay System Substrate (Promega). The luminescence level was measured, and the relative light units (RLUs) of luciferase were determined using a GloMax 96 luminometer (Promega). To analyze the efficiency of viral genome packaging, the RLUs acquired from cells infected with VLPs were divided by the RLUs acquired from cells transfected with corresponding minigenome plasmids. Bar graphs were drawn using GraphPad Prism 7 software (GraphPad Software, Inc.) and statistically analyzed using an unpaired two-tailed *t*-test.

### 2.8. Viral Growth Kinetics

Viral growth kinetics of wtLCMV, recombinant non-mutated wildtype LCMV (rwtLCMV), and rLCMV with various mutations in Vero or A549 cells were analyzed. Briefly, confluent monolayers of Vero or A549 cells cultured in 12-well plates were infected with each LCMV at an MOI of 0.001 per cell. Cells were washed 3 times with DMEM-2FBS after a one-hour adsorption period at 37 °C, and 1 mL DMEM-2FBS was added to each well. Supernatant samples were collected at 0, 24, 48, 72, 96, and 120 h postinfection. The supernatants were centrifuged at 8000× *g* for 5 min to remove cell debris and were stored at −80 °C until the infectious dose was measured using a viral immunofocus assay. Viral growth curves of LCMVs were drawn using GraphPad Prism 7 software and statistically analyzed using two-way ANOVA.

### 2.9. Viral Genetic Stability Test

Confluent monolayers of Vero cells cultured in 12-well plates were infected with each LCMV at an MOI of 0.001 per cell, and infected cells were cultured at 37 °C for 3 days. Then, 200 µL of the supernatant was serially passaged to Vero cells freshly prepared in 3- or 4-day intervals. Serial passage was conducted 10 times; RNA of each rLCMV was extracted from the 10-times passaged supernatant; and viral genome sequences of all rLCMVs were analyzed by Sanger Sequencing of viral cDNAs generated by RT-PCR method [33].

### 2.10. Animal Experiments

Animal experiments were performed in an Animal Biosafety Level 3 laboratory. All experiments were performed in accordance with the Guidelines for Animal Experimentation of National Institute of Infectious Diseases (NIID), Japan, under the approval of the Committee on Experimental Animals at NIID (No. 116038). Specific pathogen-free 7-week-old female CBA/NSlc mice and specific pathogen-free 8-week-old female DBA/1JJmsSlc mice were purchased from Japan SLC, Inc. (Shizuoka, Japan). Mice were intraperitoneally (i.p.) infected with 1.0 × 10^2^ FFU wtLCMV, rwtLCMV, mutated rLCMVs, or medium alone. Mice were monitored daily for 21 days for clinical symptoms, body weight, and survival. Mice showing more than 20% weight loss were euthanized out of ethical consideration. Blood was collected from the caudal vein of each mouse that survived to 37 days postinfection (d.p.i.) (CBA/NSlc mice) or 40 d.p.i. (DBA/1JJmsSlc mice). Following the first infection, mice that survived and showed no apparent symptoms at 40 d.p.i. were further i.p. inoculated with 1.0 × 10^3^ FFU of wtLCMV and again monitored daily for 21 days for clinical symptoms, body weight, and survival. After 21 days postinfection, mice were euthanized under isoflurane deep anesthesia. Survival curves were drawn using GraphPad Prism 7 software and statistically analyzed using the log-rank (Mantel–Cox) test. The curves of body weight changes were drawn using GraphPad Prism 7 software and statistically analyzed using multiple *t*-tests. Discovery was determined using the two-stage linear step-up produced by Benjamini, Krieger, and Yekutieli, with *Q* = 5% [34].

### 2.11. Neutralization Assay

Blood was collected from the caudal vein of each mouse at either 37 d.p.i. or 40 d.p.i. using BD Microtainer blood collection tubes (BD, Franklin Lakes, NJ, USA) and centrifuged at 8000 × *g* for 5 min. Separated plasma was inactivated by heat treatment at 56 °C for 30 min. Plasma was serially diluted with DMEM-2FBS from 1:20 to 1:160 (two-fold serial dilution) and mixed with an equal volume of DMEM-2FBS containing 40–70 FFU/50 µL wtLCMV. The mixtures were incubated for 1 h at 37 °C. Vero cells cultured in 12-well plates were inoculated with 100 µL of each mixture and cultured at 37 °C for 1 h for adsorption. The cells were overlaid with 1 mL maintenance medium (Eagle’s minimal essential medium containing 1% methylcellulose, 2 mM L-glutamine, 0.22% sodium bicarbonate, and 2% FBS). The plates were incubated at 37 °C in 5% CO_2_ for 120 h. Cells were fixed with 10% formaldehyde, permeabilized by incubation with PBS containing 0.2% Triton X-100 (Sigma-Aldrich, St. Louis, MO, USA), and then stained with anti-LCMV-WE recombinant NP immunized rabbit serum and HRP-goat anti-rabbit IgG (H+L) DS Grd (Life Technologies) [31]. The cells were then stained with Peroxidase Stain DAB Kit (Nacalai). The number of stained foci was counted as described above, and the 50% focus reduction neutralization titer (FRNT_50_) was measured. The FRNT_50_ titers were defined as the reciprocal of the serum dilution level where the focus number became less than 50% of the control, and the focus number of the wells were inoculated with the mixture of non-plasma containing DMEM-2FBS and DMEM-2FBS containing 40–70 FFU/50 µL of wtLCMV.

### 2.12. Western Blotting

Three hundred microliters of cell lysate were mixed with 100 µL 4× sample buffer solution containing 20 vol% 2-mercaptoethanol (191-13272; Wako, Osaka, Japan), and proteins were separated on a sodium dodecyl sulfate (SDS) 4–12% gradient polyacrylamide gel (Thermo Fisher Scientific, Carlsbad, CA, USA). Proteins were transferred to a polyvinylidene difluoride membrane (Bio-Rad Laboratories, Hercules, CA, USA); the membrane was then blocked by incubating in saturation buffer (PBS containing 0.25% skim milk and 0.05% Tween 20). The membrane was then incubated with primary antibodies including rabbit anti-LCMV-WE recombinant NP serum and mouse anti-α tublin monoclonal antibody (T5168; Sigma-Aldrich) [31]. The rabbit anti-LCMV-WE recombinant NP serum and the mouse anti-α tublin monoclonal antibody were diluted at 1:500 and 1:1000 in saturation buffer and used, respectively. After washing with PBS containing 0.05% Tween 20, the membrane was incubated with HRP-labeled secondary antibodies, including HRP-conjugated goat anti-rabbit IgG (H+L) DS Grd (31460; Life Technologies) or HRP-conjugated goat anti-mouse IgG(H+L) (62-6520; Thermo Fisher Scientific). The secondary antibodies were diluted at 1:1000 in saturation buffer. Protein expression was detected using Immobilon Western Chemiluminescent HRP Substrate (Millipore, Burlington, MA, USA) according to the manufacturer’s instructions. Bands were visualized using an image analyzer (LAS-3000 mini, Fujifilm, Tokyo, Japan).

## 3. Results

### 3.1. Design of Plasmids for Recombinant LCMV with Mutations in the S Segment UTRs and Prediction of Their RNA Secondary Structure

The genome structure of wtLCMV S segment is shown in Figure 1. We used the CENTROIDFOLD server to predict secondary structures of the S segment UTRs (Figure 2). In addition to the previously identified 19 nt forming the terminal panhandle structure, RNA produced from the wtLCMV genome segment (pRF-WE-SRG) also formed a double-stranded region comprising 45 and 42 nt at the 5′ and 3′ termini, respectively. In the terminal panhandle structures, nt 28–33 and 26–31 at the 5′ and 3′ termini, respectively, formed base pairs with the highest base-pairing probability in addition to the 19 base pairs at the termini. Focusing on the panhandle structure and in particular on the base pairs comprising residues 28–33 at the 5′ terminus and 26–31 at the 3′ terminus, we designed pRF-WE-SRGs with various mutations and predicted the secondary structures of their RNA products (Figure 1, Appendix A, and Table 1 and Table 2). The predicted RNA secondary structure of RNA products from pRF-WE-SRG-5UTRΔ20–40, pRF-WE-SRG-5UTRΔ41–60, pRF-WE-SRG-5UTRΔ60–77, pRF-WE-SRG-3UTRΔ20–38, and pRF-WE-SRG-3UTRΔ39–60 are shown in Appendix A and Table 3. The RNAs produced from pRF-WE-SRG-5UTRΔ41–60, pRF-WE-SRG-5UTRΔ60–77, and pRF-WE-SRG-3UTRΔ39–60 formed panhandle structures that comprised more than 36 and 33 nt at the 5′ and 3′ termini, respectively. The base pairs comprising the 28th to the 33rd nt at the 5′ terminus and the 26th to the 31st nt at the 3′ terminus showed the highest base-pairing probabilities. Only the extreme terminal panhandle of the 19 base pairs in the termini was present in the RNAs produced from pRF-WE-SRG-5UTRΔ20–40 and pRF-WE-SRG-3UTRΔ20–38.

To further investigate the effect of the base pairs comprising the 28th to the 33rd nt at the 5′ terminus and the 26th to the 31st nt at the 3′ terminus and their peripheral regions, various plasmids with mutations or deletions in the regions of interest (nt 20–40 at the 5′ terminus and 20–38 at the 3′ terminus) were generated (Figure 1). The results of RNA secondary structure prediction of the RNA products of pRF-WE-SRG-UTR-comple, pRF-WE-SRG-UTR 5-3 change, pRF-WE-SRG-Δ26–40, pRF-WE-SRG-Δ20–25, pRF-WE-SRG-Δ20–30, and pRF-WE-SRG-Δ31–40 are shown in Appendix A. A summary of the RNA secondary structure predictions is shown in Table 3. Briefly, the RNAs produced from pRF-WE-SRG-UTR-comple, pRF-WE-SRG-UTR 5-3 change, pRF-WE-SRG-Δ20–25, and pRF-WE-SRG-Δ20–30 formed panhandle structures with 5′ and 3′ termini, which were composed of the base pairs with high base-pairing probability, in addition to the 19 base pairs in the termini. However, the RNAs produced from pRF-WE-SRG-Δ26–40 and pRF-WE-SRG-Δ31–40 formed panhandle structures with 5′ and 3′ termini, but there were no base pairs with high base-pairing probability except for the extreme 19 base pairs in the termini.

### 3.2. Rescue and Characterization of Recombinant LCMVs

First, rwtLCMV was successfully generated using pRF-WE-SRG. There was no significant difference in viral growth kinetics in Vero cells between wtLCMV and rwtLCMV (*p* = 0.0723) (Figure 3a). Next, we attempted to generate rLCMVs from pRF-WE-SRG-5UTRΔ20–40, pRF-WE-SRG-5UTRΔ41–60, pRF-WE-SRG-5UTRΔ60–77, pRF-WE-SRG-3UTRΔ20–38, and pRF-WE-SRG-3UTRΔ39–60 (Table 4). Successfully generated rLCMVs included rLCMV-5UTRΔ41–60, rLCMV-5UTRΔ60–77, and rLCMV-3UTRΔ39–60 as confirmed by IFA; however, two rLCMVs (rLCMV-5UTRΔ20–40 and rLCMV-3UTRΔ20–38) could not be rescued. Recombinant viruses rLCMV-5UTRΔ41–60, rLCMV-5UTRΔ60–77, and rLCMV-3UTRΔ39–60 were propagated in Vero cells to reach titers up to 1.0 × 10^6^–10^7^ FFU/mL at 72 h postinfection (Figure 3b). No significant differences in viral growth efficiency in Vero cells were observed between rwtLCMV and rLCMV-5UTRΔ41–60 or rLCMV-5UTRΔ60–77 (*p* = 0.4698 and *p* = 0.3250, respectively); the total variation, which consists of the sum of the squares of the differences of each mean with the grand mean, were 0.06001% for rwtLCMV vs. rLCMV-5UTRΔ41–60 and 0.6165% for rwtLCMV vs. rLCMV-5UTRΔ60–77. A significant difference in viral growth efficiency in Vero cells was observed between rwtLCMV and rLCMV-3UTRΔ39–60 (*p* = 0.0001; total variation = 5.941%), but no significant difference in viral growth efficiency in Vero cells was observed between wtLCMV and rLCMV-3UTRΔ39–60 (*p* = 0.6755; total variation = 0.04217%). The viral growth capacity of rLCMV-5UTRΔ41–60 and that of rLCMV-5UTRΔ60–77 in A549 cells was equal to that of rwtLCMV (Figure 3c, *p* = 0.1292 and 0.5347; total variation = 2.945% and 0.465%, respectively). The viral growth capacity of rLCMV-3UTRΔ39–60 in A549 cells was less than that of rwtLCMV (*p* = 0.0137; total variation = 11.96%).

rLCMVs with mutations in the UTRs of interest (the 20th to the 40th nt at the 5′ terminus and the 20th to the 38th nt at the 3′ terminus) were generated from pRF-WE-SRG-UTR-comple, pRF-WE-SRG-UTR 5-3 change, pRF-WE-SRG-Δ26–40, pRF-WE-SRG-Δ20–25, pRF-WE-SRG-Δ20–30, and pRF-WE-SRG-Δ31–40 (Table 4). rLCMVs including rLCMV-UTR-comple, rLCMV-UTR 5-3 change, and rLCMV-Δ26–40 were successfully generated; although as the propagation efficiency of rLCMV-UTR 5-3 change and rLCMV-Δ26–40 was low, they were further passaged in Vero cells before use. The generation of rLCMVs rLCMV-Δ20–25, rLCMV-Δ20–30, and rLCMV-Δ31–40 was not confirmed. The viral growth capacity of rLCMV-UTR-comple, rLCMV-UTR 5-3 change, and rLCMV-Δ26–40 in Vero cells was significantly less than that of rwtLCMV (Figure 3d, *p* = 0.0008, 0.0003, and 0.0003; total variation = 22.55%, 31.86%, and 30.51%, respectively).

The focus morphologies of wtLCMV and rLCMVs generated are shown in Figure 3e. The size of foci was concordant with viral growth capacity features. The focus sizes of rLCMV-5UTRΔ41–60, rLCMV-5UTRΔ60–77, and rLCMV-3UTRΔ39–60 were equivalent to those of wtLCMV and rwtLCMV. The focus sizes of rLCMV-UTR-comple and rLCMV-Δ26–40 were smaller than those of wtLCMV and rwtLCMV, while the focus size of rLCMV-UTR 5-3 change was the smallest.

### 3.3. Genetic Stability of rLCMVs

To investigate the genetic stability of rLCMVs, the supernatants of rLCMV-infected cells were serially passaged in Vero cells 10 times and the genome sequence of the 10-times passaged (P10) rLCMVs were compared with the original (P0) rLCMVs. The results of genome sequence analysis of P10 rLCMVs are summarized in Table 5. Briefly, rLCMV-5UTRΔ60–77, rLCMV-UTR-comple, rLCMV-UTR 5-3 change, and rLCMV-Δ26–40 obtained mutations through serial passage. Most mutations accumulated in S segment UTRs or L protein ORF. One mutation which caused synonymous amino acid change was found in GPC of rLCMV-UTR 5-3 change. No mutations were observed in nucleic acid sequences of rwtLCMV, rLCMV-5UTRΔ41–60, and rLCMV-3UTRΔ39–60.

### 3.4. Pathogenicity of rLCMVs in Mice

To investigate the virulence of the rLCMVs (rwtLCMV, rLCMV-5UTRΔ41–60, rLCMV-5UTRΔ60–77, rLCMV-3UTRΔ39–60, rLCMV-UTR-comple, rLCMV-UTR 5-3 change, and rLCMV-Δ26–40), CBA/NSlc and DBA/1JJmsSlc mice were infected i.p. with wtLCMV or each of the generated rLCMVs at the dose of 1.0 × 10^2^ FFU/mouse.

CBA/NSlc mice infected with wtLCMV (wtLCMV-CBA/NSlc mice), rwtLCMV (rwtLCMV-CBA/NSlc mice), rLCMV-5UTRΔ41–60 (rLCMV-5UTRΔ41–60-CBA/NSlc mice), and rLCMV-5UTRΔ60–77 (rLCMV-5UTRΔ60–77-CBA/NSlc mice) showed symptoms such as ruffled fur and limb tremors at 7 d.p.i., and weight loss at 8 d.p.i. (Figure 4a). The survival rate of each mouse group is shown in Figure 4b and Table 4. Significant differences in survival curves were not observed between rwtLCMV and rLCMV-5UTRΔ60–77 (*p* = 0.3997) and between rwtLCMV and rLCMV-5UTRΔ41–60 (*p* = 0.3148). Hazard ratios for rwtLCMV/rLCMV-5UTRΔ60–77 and rwtLCMV/rLCMV-5UTRΔ41–60 were 1.517 and 2.266, respectively. However, CBA/NSlc mice infected with either rLCMV-3UTRΔ39–60 (rLCMV-3UTRΔ39–60-CBA/NSlc mice), rLCMV-UTR-comple (rLCMV-UTR-comple-CBA/NSlc mice), rLCMV-UTR 5-3 change (rLCMV-UTR 5-3 change-CBA/NSlc mice), or rLCMV-Δ26–40 (rLCMV-Δ26–40-CBA/NSlc mice) were symptom free and survived. Significant differences in survival curves were observed between rwtLCMV and either rLCMV-3UTRΔ39–60, rLCMV-UTR-comple, rLCMV-UTR 5-3 change, or rLCMV-Δ26–40 (*p* = 0.0133, respectively) (Figure 4a,b, and Table 4).

All DBA/1JJmsSlc mice infected with wtLCMV (wtLCMV-DBA/1JJmsSlc mice) or rwtLCMV (rwtLCMV-DBA/1JJmsSlc mice) showed symptoms such as ruffled fur, limb tremors, and weight loss at 5 d.p.i., and approximately half the wtLCMV-DBA/1JJmsSlc mice and rwtLCMV-DBA/1JJmsSlc mice died within 13 d.p.i. (Figure 4c,d and Table 4). DBA/1JJmsSlc mice infected with rLCMV-5UTRΔ41–60 (rLCMV-5UTRΔ41–60-DBA/1JJmsSlc mice), rLCMV-5UTRΔ60–77 (rLCMV-5UTRΔ60–77-DBA/1JJmsSlc mice), or rLCMV-3UTRΔ39–60 (rLCMV-3UTRΔ39–60-DBA/1JJmsSlc mice) showed symptoms by 7 d.p.i. However, limb tremors were not observed and ruffled fur was mild. All rLCMV-5UTRΔ41–60-DBA/1JJmsSlc mice and rLCMV-5UTRΔ60–77-DBA/1JJmsSlc mice lost weight but survived. Although one rLCMV-3UTRΔ39–60-DBA/1JJmsSlc mouse lost weight and died at 8 d.p.i., the other 4 mice did not lose weight and survived. Significant differences in survival curves were not observed between rwtLCMV-DBA/1JJmsSlc and rLCMV-3UTRΔ39–60-DBA/1JJmsSlc (*p* = 0.2194). The hazard ratio for rLCMV/rLCMV-3UTRΔ39–60 was 3.603. All DBA/1JJmsSlc mice infected with either rLCMV-UTR-comple (rLCMV-UTR-comple-DBA/1JJmsSlc mice), rLCMV-UTR 5-3 change (rLCMV-UTR 5-3 change-DBA/1JJmsSlc mice), or rLCMV-Δ26–40 (rLCMV-Δ26–40-DBA/1JJmsSlc mice) showed no clinical signs and survived (Figure 4c,d and Table 4).

### 3.5. Acquired Immunity against LCMV in Mice Induced by Infection With rLCMVs

Neutralization titers against LCMV in sera collected from CBA/NSlc mice at 37 days after first infection and from DBA/1JJmsSlc mice at 40 days after the first infection were evaluated. The 53 sera specimens collected from rLCMV-infected mice all showed negative reactions in the focus reduction neutralization test with the exception of one specimen from an rLCMV-3UTRΔ39–60-CBA/NSlc mouse. The FRNT_50_ of the positive sample was 40.

Mice that survived inoculations with rLCMVs were further i.p. infected with 1.0 × 10^3^ FFU of wtLCMV 40 days after the first inoculation. All CBA/NSlc mice previously inoculated with medium alone (the control) died within 11 d.p.i. Conversely, all rLCMV-5UTRΔ41–60-, rLCMV-5UTRΔ60–77-, rLCMV-3UTRΔ39–60-, rLCMV-UTR-comple-, rLCMV-UTR 5-3 change-, and rLCMV-Δ26–40-CBA/NSlc mice survived the viral challenge. Furthermore, they were all symptom-free (Figure 5a,b and Table 4).

All control DBA/1JJmsSlc mice died within 8 d.p.i. Conversely, all rLCMV-5UTRΔ41–60-, rLCMV-5UTRΔ60–77-, rLCMV-3UTRΔ39–60-, rLCMV-UTR-comple-, rLCMV-UTR 5-3 change-, and rLCMV-Δ26–40- DBA/1JJmsSlc mice (with the exception of one rLCMV-Δ26–40-DBA/1JJmsSlc mouse) survived the secondary viral challenge as well (Figure 5c,d and Table 4). None of the rLCMV-5UTRΔ41–60-, rLCMV-5UTRΔ60–77-, or rLCMV-3UTRΔ39–60- DBA/1JJmsSlc mice showed any clinical symptoms. One of the 5 rLCMV-UTR-comple-DBA/1JJmsSlc mice showed ruffled fur and weight loss at 6 d.p.i. Several rLCMV-UTR 5-3 change- and rLCMV-Δ26–40- DBA/1JJmsSlc mice showed weight loss 2 d.p.i.; all rLCMV-UTR 5-3 change- and rLCMV-Δ26–40- DBA/1JJmsSlc mice showed ruffled fur 5 d.p.i.; and one rLCMV-Δ26–40-DBA/1JJmsSlc mouse died 8 d.p.i.

### 3.6. Minigenome Assay and VLP Assay

The effect of various mutations or deletions in the UTRs on viral genome transcription and replication was assessed with the established minigenome assay. The function of UTRs with mutated SMGs was also analyzed. To construct SMG-luc or SMG-GFP, cDNA fragments containing the S 5′ UTR, S IGR, GFP, or renilla luciferase ORFs in an antisense orientation with respect to the 5′ UTR, and the S 3′ UTR were cloned between the murine pol I promoter and terminator of the pRF vector system (Figure 6a). Replication, transcription, and translation of SMG-luc and SMG-GFP are shown in Figure 6a. The expression of luciferase or GFP was only observed in BHK-21 cells when these were co-transfected with SMG-luc (or SMG-GFP) and both pC-NP and pC-L (Figure 6b and Appendix A). The signal-to-noise ratio for the established minigenome assay was 7.9 × 10^2^.

We generated various mutated SMGs in which the mutations were equivalent to the mutations introduced to pRF-WE-SRGs (Table 1 and Table 2). Detailed information relating to the mutated SMG-luc or -GFP is described in the Materials and Methods section. We evaluated the efficiency of genome transcription and replication of the RNAs derived from SMG-luc (or -GFP), SMG-5UTRΔ20–40-luc (or -GFP), SMG-5UTRΔ41–60-luc (or -GFP), SMG-5UTRΔ60–77-luc (or -GFP), SMG-3UTRΔ20–38-luc (or -GFP), SMG-3UTRΔ39–60-luc (or -GFP), and SMG-UTR-comple-luc (Figure 7a and Appendix A). The expression of LCMV NP and α tublin is shown in Figure 7b. The expression of LCMV NP was indistinguishable from that observed in pC-NP transfected samples. The expression of minigenome-derived reporter genes was hampered by the co-expression of GPC and Z protein as previously reported (Appendix A, SMG-GFP versus SMG-GFP, no VLPs) [11]. Viral genome transcription and replication were not observed in cells transfected with SMG-5UTRΔ20–40-luc or SMG-5UTRΔ20–40-GFP. The efficiency of viral genome transcription and replication in the cells transfected with SMG-3UTRΔ20–38-luc or SMG-UTR-comple-luc was significantly lower compared with cells transfected with SMG-luc. Luciferase expression in cells transfected with SMG-5UTRΔ41–60-luc, SMG-5UTRΔ60–77-luc, or SMG-3UTRΔ39–60-luc was equivalent to that of cells transfected with SMG-luc (Figure 7a).

We also evaluated the packaging efficiency of the viral genome RNA analogs into VLPs (Figure 7c and Appendix A). The luciferase expression in cells infected with VLPs, which encapsulated RNA products derived from SMG, SMG-5UTRΔ41–60-luc, SMG-5UTRΔ60–77-luc, or SMG-3UTRΔ39–60-luc, were approximately equal to each other. Luciferase expression in the cells infected with VLPs that encapsulated RNA products derived from SMG-UTR-comple-luc was significantly lower than that in the cells infected with VLPs that encapsulated RNA products derived from SMG-luc but significantly higher than that of the cells treated with a supernatant derived from cells transfected with SMG-luc without VLP expression plasmids (pC-Z and pC-GPC) (*p* = 0.0003) (Figure 7c).

## 4. Discussion

Here, we described an LCMV-WE reverse genetics system and a minigenome system (Figure 3a and Figure 6). A polymerase-I-driven LCMV-ARM reverse genetics system has been previously reported with analyses of LCMV-ARM [13,15]. The identities of the nucleotide sequences between the S and L segments of LCMV-WE and LCMV-ARM are 85% and 82%, respectively, and several reports have described characteristic differences between these strains [19,22,26,27]. Our LCMV-WE reverse genetics system will enable us to elucidate the mechanistic significance of these differences from a virological perspective in future studies. For instance, it became possible to generate chimeric LCMV between LCMV-WE and LCMV-ARM and to determine which viral proteins were responsible for the pathogenic difference in mice between LCMV-WE and LCMV-ARM. Furthermore, the infection of NHPs with rLCMVs might provide further understanding of the arenavirus-mediated pathogenic mechanisms associated with viral hemorrhagic fever [23,24,25,26,27].

In this study, we focused on the S segment UTRs and performed RNA secondary structure prediction for these (Figure 2). The 28th to 33rd nt in the 5′ terminus and the 26th to 31st nt in the 3′ terminus formed base pairs with the highest base-pairing probability (Table 3). The base pairs comprising the 20th to the 40th nt in the 5′ terminus and the 20th to the 38th nt in the 3′ terminus, which included nt 28–33 in the 5′ terminus and 26–31 in the 3′ terminus, greatly affected viral propagation (Appendix A, Figure 3, and Table 4). The successful generation of rLCMV-UTR-comple, rLCMV-UTR 5-3 change, and rLCMV-Δ26–40 and the failure to generate rLCMV-Δ20–25, rLCMV-Δ20–30, and rLCMV-Δ31–40 suggested that both the conformation of the panhandle structure and the specific nucleotide sequences of these base pairs may be important for the recognition by the L protein. However, further studies will be needed to confirm or refute this hypothesis. The predicted RNA secondary structure of the rLCMV-Δ26–40 S segment lacked base pairs with high base-pairing probability with the exception of the 19 base pairs in the extreme termini. The reason why rLCMV-Δ26–40 successfully propagated was not clarified in this study, but the conservation of base pairs comprising nt 20–25 in both termini and the RNA conformation may be advantageous to propagation.

The viral growth capacity and focus size of rLCMV-5UTRΔ41–60, rLCMV-5UTRΔ60–77, and rLCMV-3UTRΔ39–60 in Vero cells were equivalent or similar to those of wtLCMV and rwtLCMV (Figure 3b,e and Table 4). This suggests that the conservation of the base pairs comprising the 20–40 nt region in the 5′ terminus and the 20–38 nt region in the 3′ terminus was not essential but did have a considerable effect on viral propagation, though the other UTRs (nt 41–77 in the 5′ terminus and nt 39–60 in the 3′ terminus) did not affect viral propagation in vitro.

The mutations or deletions in UTRs of the S segment of LCMV attenuated virulence in vivo (Figure 4 and Table 4). In particular, mutations in the base pairs of the 20–40 nt region in the 5′ terminus and the 20–38 nt region in the 3′ terminus (rLCMV-UTR-comple, rLCMV-UTR 5-3 change, and rLCMV-Δ26–40) caused major in vivo attenuation of LCMV. Low efficiency of viral propagation in vitro may lead to viral attenuation in vivo. Significant differences in the survival curves were observed between rwtLCMV-CBA/NSlc mice and rLCMV-3UTRΔ39–60-CBA/NSlc mice, between rwtLCMV-DBA/1JJmsSlc mice and rLCMV-5UTRΔ41–60-DBA/1JJmsSlc mice, and between rwtLCMV-DBA/1JJmsSlc mice and rLCMV-5UTRΔ60–77-DBA/1JJmsSlc mice. There were no significant differences in the survival curves between rwtLCMV-CBA/NSlc mice and rLCMV-5UTRΔ60–77-CBA/NSlc mice or rLCMV-5UTRΔ41–60-CBA/NSlc mice and between rwtLCMV-DBA/1JJmsSlc mice and rLCMV-3UTRΔ39–60-DBA/1JJmsSlc mice. However, the hazard ratios obtained in this study suggested that rLCMV-5UTRΔ41–60, rLCMV-5UTRΔ60–77, and rLCMV-3UTRΔ39–60 possessed less virulence compared with wtLCMV and rwtLCMV. Reports have suggested that UTRs in other viruses were involved in virulence, where the variable region of the 3′ UTR of the tick-borne encephalitis virus genome was associated with virulence in mice [35,36]. Additionally, UTRs of the picornavirus genome were reported to affect viral infection and host innate immunity [37]. In our study, the role of nt 41–77 in the 5′ UTR and nt 39–60 in the 3′ UTR was not elucidated. However, the results of the mouse experiments indicate these UTRs may affect host innate immunity or other factors in vivo. CBA/NSlc mice are widely used in infectious disease and immunity research [38,39,40]. A single amino acid mutation in Bruton’s tyrosine kinase reduces the function of B cells in CBA/NSlc mice [41,42]. On the other hand, DBA/1JJmsSlc mice are used in arthritis and immunity research and their B cell function is competent [43,44,45]. Although one of five rLCMV-3UTRΔ39–60-DBA/1JJmsSlc mice lost body weight and died, none of the rLCMV-3UTRΔ39–60-CBA/NSlc mice showed any apparent symptoms and survived the first infection. These results suggest the 39–60 nt region in the 3′ UTR may affect innate immunity; the low growth efficiency of rLCMV-3UTRΔ39–60 in A549 cells, in which the type I interferon pathway is intact, supports this hypothesis (Figure 3c). However, further investigation is required to make a firm conclusion about this.

The results of the secondary infection with a lethal dose of wtLCMV in mice that survived first infection suggest that protective immunity was induced in these mice (Figure 5 and Table 4). rLCMV-UTR-comple-, rLCMV-UTR 5-3 change-, and rLCMV-Δ26–40- DBA/1JJmsSlc mice developed symptoms following the secondary infection, and one mouse died at 8 d.p.i. Conversely, the rLCMV-5UTRΔ41–60-, rLCMV-5UTRΔ60–77-, or rLCMV-3UTRΔ39–60-DBA/1JJmsSlc mice lacked any clinical symptoms. These results indicate that mice infected with rLCMVs that propagate in a similar way to wtLCMV induced higher protective immunity than mice infected with rLCMVs that have a low capacity for propagation. Taking into consideration the reduced function of B cells in CBA/NSlc mice, neutralizing antibodies against LCMV were still not induced in most mice. This is consistent with previous studies, where the generation of anti-LCMV neutralizing antibodies was not detectable between 60 and 120 d.p.i. following LCMV-WE infection in mice and where CD8^+^ T-cell-mediated cytotoxicity played a key role in LCMV-WE infection [46].

Deletions in the UTRs of the Bunyamwera virus (BUNV) can attenuate viral growth properties, but these were recovered following BUNV serial passage in vitro [47]. Amino acid changes were found in the C-terminal domain of the BUNV L protein after the serial passage, indicating that these changes may be involved in the evolution of the L polymerase, which enabled more efficient recognition of the deleted UTRs. In our study, deletions or other mutations in the base pairs comprising the 20–40 nt region in the 5′ terminus and the 20–38 nt region in the 3′ terminus greatly hampered viral genome transcription and replication (Figure 7a and Appendix A). This may be the cause of the non-production of rLCMVs derived from pRF-WE-SRG-5UTRΔ20–40 and pRF-WE-SRG-3UTRΔ20–38. These results supported the notion that the panhandle structure composed of the base pairs targeted in these regions was involved in the recognition site of the L protein. In the genetic stability test, amino acid changes were mainly accumulated in the RdRp region and PB2-like region of the L protein and in the UTRs of the rLCMVs that showed low growth efficiency in Vero cells (rLCMV-UTR-comple, rLCMV-UTR 5-3 change, and rLCMV-Δ26–40). Therefore, the L protein of these rLCMVs may have adapted its conformation to the mutated UTRs as indicated in the BUNV research. Deletions in the other UTRs (the 5′ UTR 41–60 region, the 5′ UTR 60–77 region, and the 3′ UTR 39–60 region) did not affect viral genome transcription and replication efficiency in the minigenome assay or viral genome packaging efficiency in the VLP assay (Figure 7 and Appendix A). These results suggest that UTRs, except for those base pairs composed of 40 nt in the 5′ terminus and 38 nt in the 3′ terminus, were not involved in viral genome transcription, replication, or packaging. Furthermore, the minigenome assay data suggest a low possibility for the conformational adaptation of L protein to the mutated UTRs in rLCMV-5UTRΔ41–60, rLCMV-5UTRΔ60–77, and rLCMV-3UTRΔ39–60. In fact, no genetic mutations were observed in rLCMV-5UTRΔ41–60 and rLCMV-3UTRΔ20–38 and only one mutation was found in rLCMV-5UTRΔ60–77 after serial passage.

Although luciferase expression levels from SMG-3UTRΔ20–38-luc were higher than those from SMG-UTR-comple-luc, the packaging efficiency of the RNA products from SMG-UTR-comple-luc was significantly higher compared with that of SMG-3UTRΔ20–38-luc (Figure 7a,c). These results suggest that RNA products derived from SMG-UTR-comple-luc were certainly packaged into VLPs and carried to the next cells but at low efficiency and that the sequence and/or conformation of UTRs also affected viral genome packaging efficiency.

In summary, we found that, in addition to the 19 nucleotide base pairs in both termini of the S segment, the base pairs between the 20–40 nt in the 5′ terminus and the 20–38 nt in the 3′ terminus of the S segment were predicted to form panhandle structures with high base-pairing probabilities. These regions affected viral propagation as they were heavily involved in viral genome transcription and replication in terms of their conformation and/or nucleotide sequence. Furthermore, our findings suggest that the other UTRs were involved in viral pathogenicity in vivo, though they did not affect the efficiency of viral genome transcription, replication, and packaging. The mechanism of attenuation of the virus in vivo remains to be determined, but the attenuation of LCMV without amino acid changes in component proteins might help us develop new vaccines.

## 5. Conclusions

In addition to the 19 nucleotide base pairs in both termini of the LCMV S segment, the base pairs between the 20–40 nt in the 5′ terminus and the 20–38 nt in the 3′ terminus of LCMV S segment were predicted to form panhandle structures and were heavily involved in viral genome transcription and replication. Furthermore, the other LCMV S segment UTRs were involved in viral pathogenicity in vivo, though they did not affect the efficiency of viral genome transcription, replication, and packaging. The mechanism of attenuation of the virus in vivo remains to be determined, but the attenuation of LCMV might help us develop new vaccines.

## Figures and Tables

**Figure 1 viruses-12-00896-f001:**
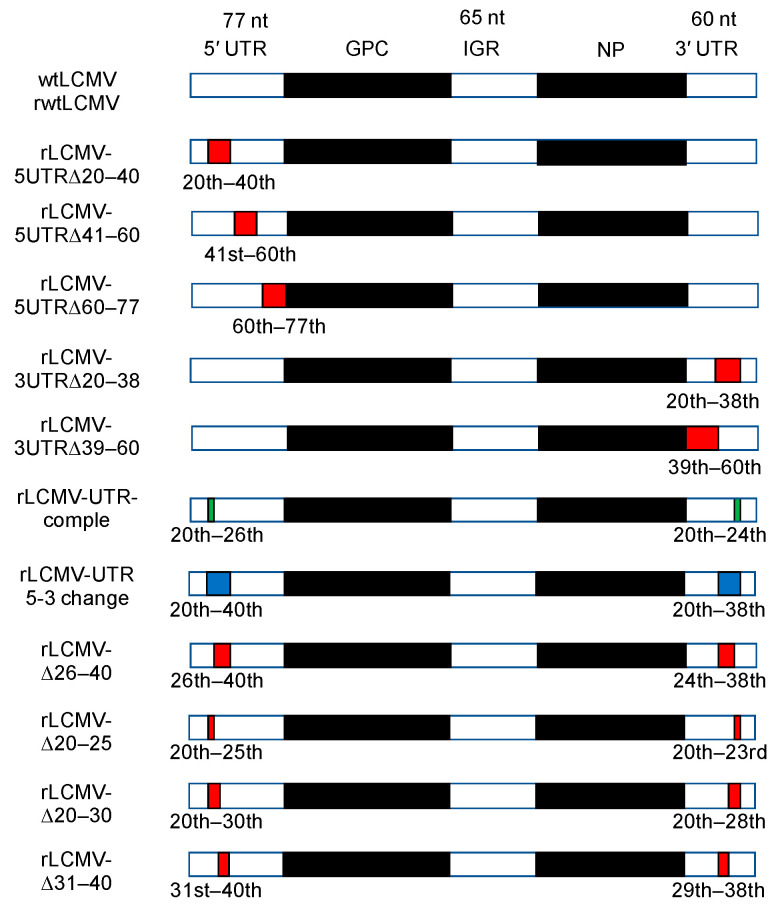
Schematics of the S segment of wildtype lymphocytic choriomeningitis virus (wtLCMV) and various mutated recombinant LCMVs (rLCMVs). The number above the region name (5′ untranslated region (UTR), intergenic region (IGR), and 3′ UTR) indicates the number of the nucleic acid in these regions. Red colored regions and the ordinal numbers described below indicate the deleted regions of S segment. Green colored regions and the ordinal numbers described below indicate the region which have complementary nucleic sequences of the relevant ordinal numbers. Blue colored regions and the ordinal numbers indicate the regions in which nucleic acid sequence of 5′ and 3′ termini are replaced by each other. All ordinal numbers are counted from either 5′ or 3′ termini, whichever is the nearest. The S segment structures of each virus names described on the left side of the schematics correspond to each schematic shown in the right side. The details on how each LCMVs were generated are described in the Plasmids subsection and in the Transfection, Minigenome, and Reverse Genetics System subsection in the Methods section.

**Figure 2 viruses-12-00896-f002:**
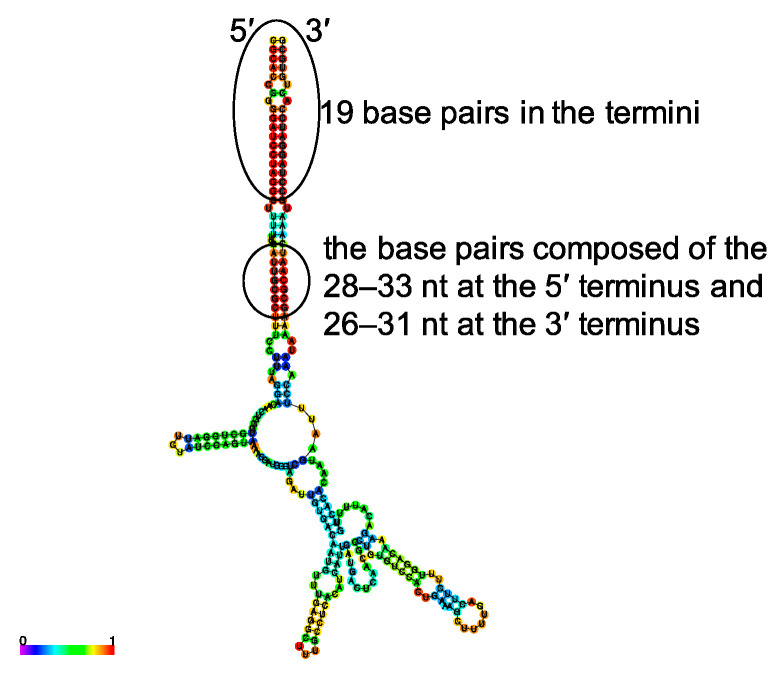
Prediction of RNA secondary structures of LCMV strain WE (LCMV-WE) S segment UTR: The predicted RNA secondary structure of LCMV-WE S segment UTR derived from pRF-WE-SRG. The location of the 19 base pairs in the termini and the base pairs comprising the 28th–33rd nt in the 5′ terminus and the 26th–31st nt in the 3′ terminus are shown. RNA sequences of LCMV-WE S segment genome 5′-terminal and 3′-terminal UTRs and 50 nt of open reading frame (ORF) regional RNA sequences that were directly downstream of the UTRs were linked, sent to the CENTROIDFOLD server, and analyzed using the CONTRAfold model (weight of base pairs: 2^2^). Each predicted base pair is colored with heat-color gradation from blue to red, corresponding to the base-pairing probability from 0 to 1. The labels “5” and “3” indicate RNA 5′ and 3′ termini, respectively.

**Figure 3 viruses-12-00896-f003:**
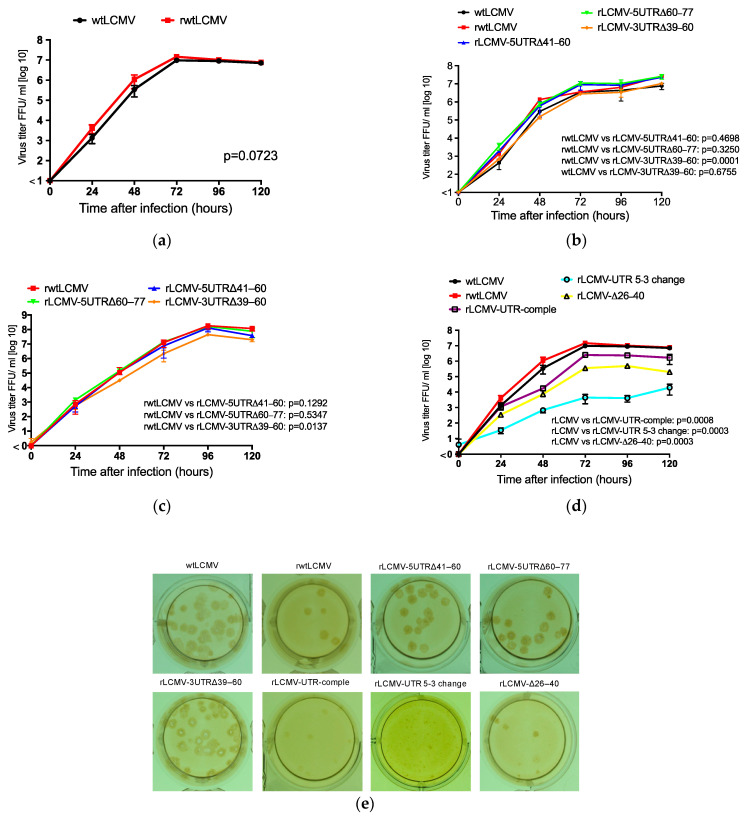
Viral growth properties in Vero and A549 cells: Confluent monolayers of Vero and A549 cells were infected with wtLCMV or rLCMVs at a multiplicity of infection (MOI) of 0.001 per cell. Cells were washed three times with medium after a one-hour adsorption period, and 1 mL medium was added to each well. Supernatant samples were collected at 0, 24, 48, 72, 96, and 120 h postinfection. The supernatants were centrifuged at 8000× *g* for 5 min to remove cell debris and were stored at −80 °C. The infectious dose was measured using a viral immunofocus assay. Viral growth curves of LCMVs were statistically analyzed using two-way ANOVA. (**a**) Viral growth properties of wildtype LCMV (wtLCMV) and recombinant non-mutated wildtype LCMV (rwtLCMV) in Vero cells were compared. (**b**) Viral growth kinetics of rLCMV-5UTRΔ41–60, rLCMV-5UTRΔ60–77, and rLCMV-3UTRΔ39–60 in Vero cells were compared with wtLCMV and rwtLCMV. (**c**) Viral growth kinetics of rLCMV-5UTRΔ41–60, rLCMV-5UTRΔ60–77, and rLCMV-3UTRΔ39–60 in A549 cells were compared with rwtLCMV. (**d**) Viral growth kinetics of rLCMV-UTR-comple, rLCMV-UTR 5-3 change, and rLCMV-Δ26–40 in Vero cells were compared with wtLCMV and rwtLCMV. (**e**) Focus morphology of wtLCMV and rLCMVs in Vero cells: The focus morphologies of wtLCMV and rLCMVs generated in this study are shown. Error bars indicate standard deviations. All experiments shown were conducted in triplicate.

**Figure 4 viruses-12-00896-f004:**
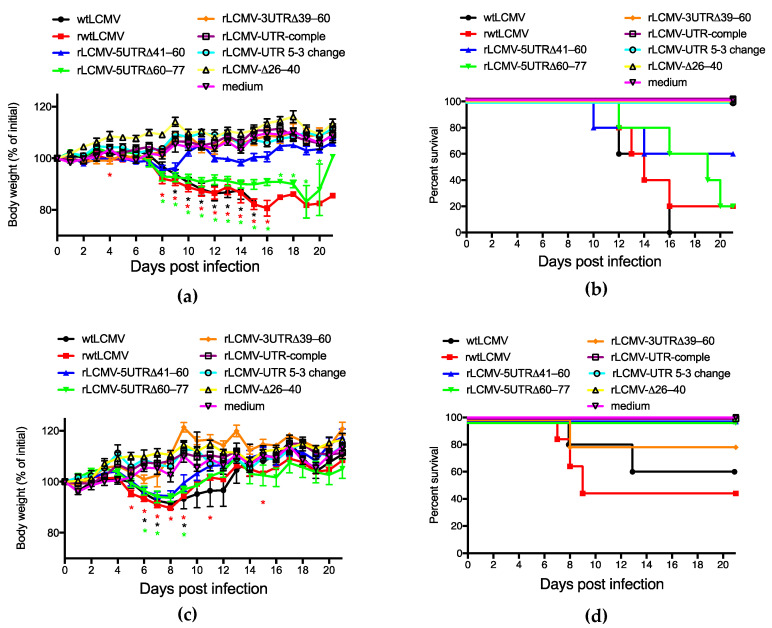
Virulence of wtLCMV and rLCMVs in CBA/NSlc mice and DBA/1JJmsSlc mice: (**a**,**b**) Changes in body weight and survival rate of CBA/NSlc mice. Seven-week-old female CBA/NSlc mice (5 mice per group) were intraperitoneally (i.p.) infected with 1.0 × 10^2^ focus-forming units (FFU) of wtLCMV, rwtLCMV, or various mutated rLCMVs. (**c**,**d**) Changes in body weight and survival rate of DBA/1JJmsSlc mice: Eight-week-old female DBA/1JJmsSlc (5 mice per group) were i.p. infected with 1.0 × 10^2^ FFU of wtLCMV or various mutated rLCMVs. Error bars in Figure 4a,c indicate standard errors of the mean. Asterisks indicate that significant differences were observed between the mean body weight of mice infected with medium alone and that of mice infected with LCMVs, displayed in the same colors.

**Figure 5 viruses-12-00896-f005:**
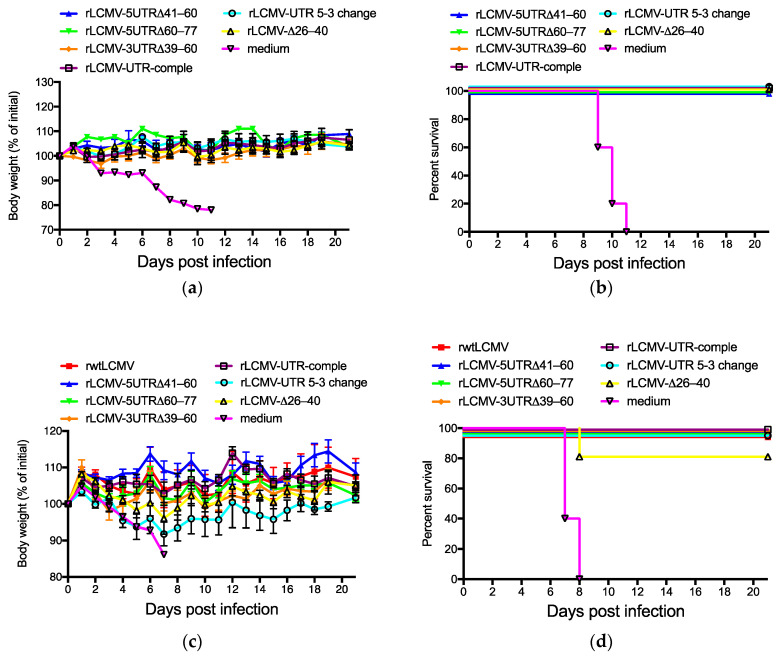
Mice previously infected with rLCMVs were given a lethal dose infection of wtLCMV. Mice that survived inoculation with rLCMVs were further infected intraperitoneally (i.p.) with 1.0 × 10^3^ FFU of wtLCMV 40 days after the first inoculation. (**a**,**b**) Changes in body weight and survival rate of CBA/NSlc mice: CBA/NSlc mice previously inoculated with rLCMV-5UTRΔ41–60 (3 mice per group), rLCMV-5UTRΔ60–77 (1 mouse per group), rLCMV-3UTRΔ39–60 (5 mice per group), rLCMV-UTR-comple (5 mice per group), rLCMV-UTR 5-3 change (5 mice per group), rLCMV-Δ26–40 (5 mice per group), or medium (5 mice per group) were used. (**c**,**d**) Changes in body weight and survival rate of DBA/1JJmsSlc mice: DBA/1JJmsSlc mice previously inoculated with rwtLCMV (3 mice per group), rLCMV-5UTRΔ41–60 (5 mice per group), rLCMV-5UTRΔ60–77 (5 mouse per group), rLCMV-3UTRΔ39–60 (4 mice per group), rLCMV-UTR-comple (5 mice per group), rLCMV-UTR 5-3 change (5 mice per group), rLCMV-Δ26–40 (5 mice per group), or medium (5 mice per group) were used. Error bars in Figure 5a,c indicate standard errors of the mean.

**Figure 6 viruses-12-00896-f006:**
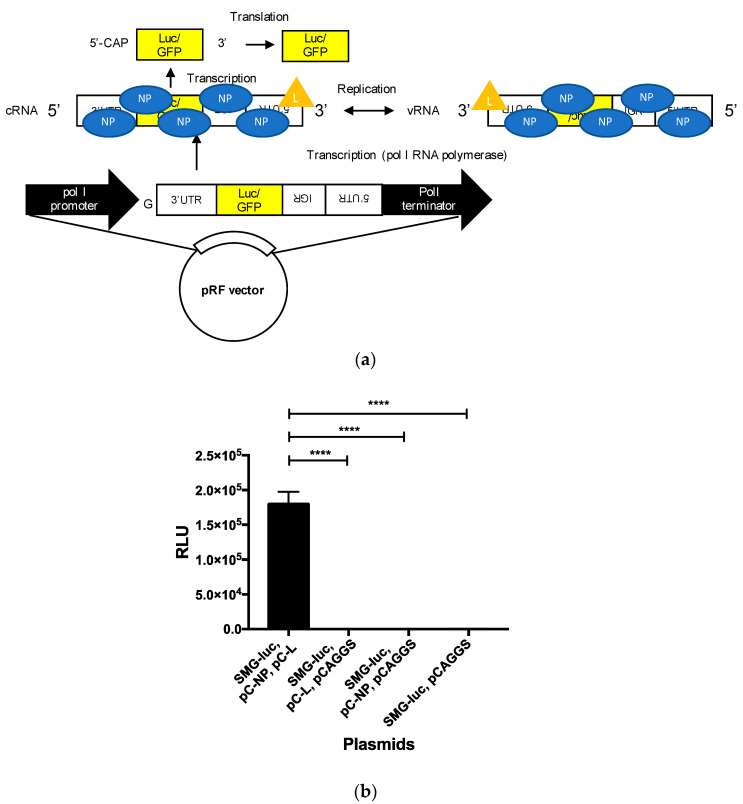
Establishment of a minigenome system for LCMV strain WE (LCMV-WE): (**a**) a schematic diagram of SMG-luc or -GFP and its replication, transcription, and translation processes. A cDNA fragment containing the S 5′ UTR, S IGR, renilla luciferase, or GFP ORFs in an antisense orientation with respect to the 5′ UTR and the S 3′ UTR were cloned between the murine pol I promoter and the terminator of a pRF vector. An additional G residue was inserted between the murine pol I promoter and the viral genome sequence. The SMG plasmids are transcribed by the polymerase I RNA polymerase to generate cRNA in BHK cells. The cRNAs are then encapsidated with nucleoprotein (NP). vRNAs are produced in the presence of L protein (L). The encapsidated cRNA is transcribed into reporter gene mRNA. Finally, transcribed mRNA is translated by host-cell machinery to produce reporter proteins (luciferase or GFP). (**b**) BHK-21 cells were transfected with minigenome plasmids SMG-luc, either pC-NP or pCAGGS, either pC-L or pCAGGS, and pC-Fluc. The transfected cells were incubated for 2 days at 37 °C, and then renilla luciferase activity was measured using the Renilla Luciferase Assay System and firefly luciferase activity was measured using the Bright-Glo Luciferase Assay System. (**** *p* < 0.0001) Error bars indicate standard deviations. The experiments were performed three times independently.

**Figure 7 viruses-12-00896-f007:**
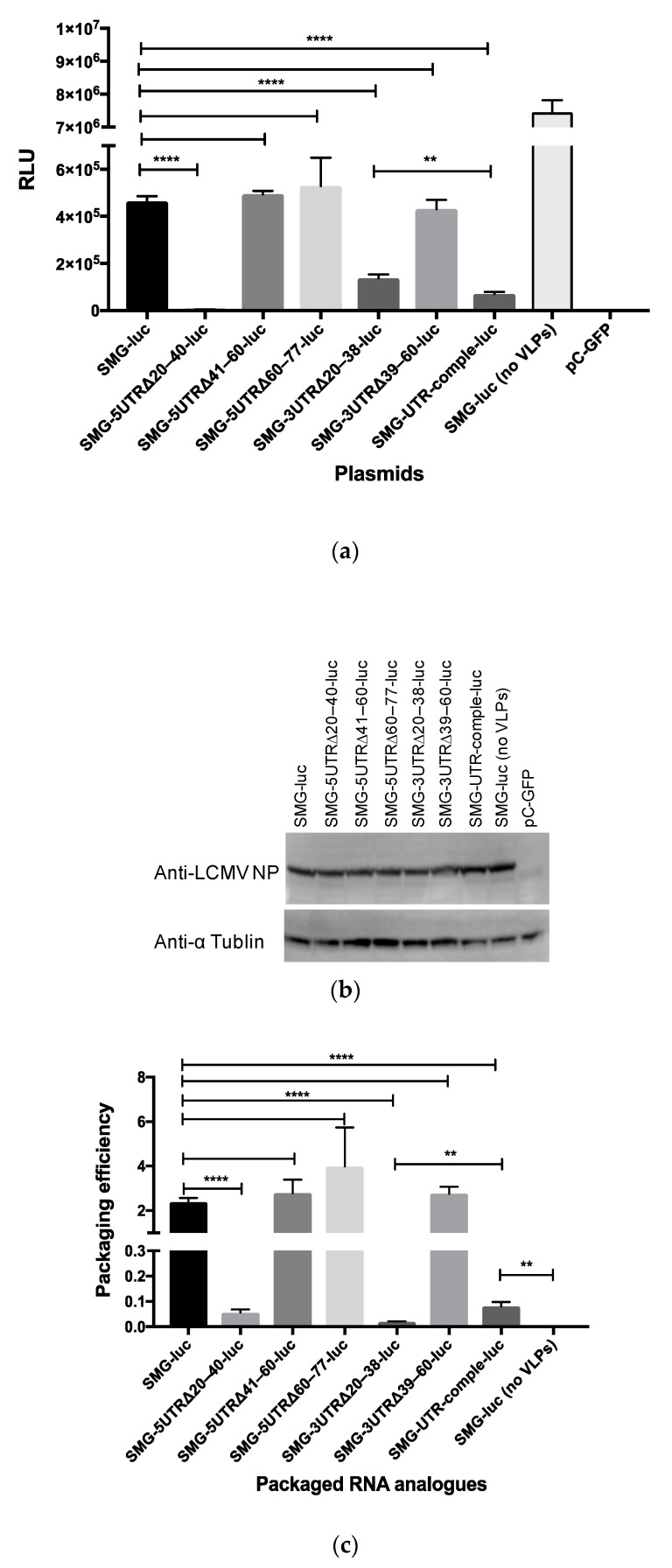
Evaluation of the efficiency of viral genome transcription, replication, and packaging in virus-like particles (VLPs): (**a**) Viral genome transcription and replication were assessed in BHK-21 cells transfected with SMG-luc, SMG-5UTRΔ20–40-luc, SMG-5UTRΔ41–60-luc, SMG-5UTRΔ60–77-luc, SMG-3UTRΔ20–38-luc, SMG-3UTRΔ39–60-luc, or SMG-UTR-comple-luc under conditions of the co-expression of NP, L protein, GPC, and Z protein. BHK-21 cells were transfected with each mutated SMG-luc, pC-NP, pC-L, pC-Z, and pC-GPC. As a background control for the rescue of LCMV RNA analogs into VLPs, cells transfected with SMG-luc, pC-NP, and pC-L were used (SMG-luc (no VLPs)). After incubation for 48 h at 37 °C with 5% CO2, luciferase expression was examined. (**b**) The expressions of LCMV NP and α tublin in transfected cells were confirmed by Western blot analysis. (**c**) Packaging of viral genome RNA analogs derived from SMG-luc, SMG-5UTRΔ20–40-luc, SMG-5UTRΔ41–60-luc, SMG-5UTRΔ60–77-luc, SMG-3UTRΔ39–60-luc, or SMG-UTR-comple-luc into VLPs were assessed. Supernatants from each well of Figure 7a were harvested and used to infect fresh monolayers of BHK-21 cells, which were then incubated for 4 h at 37 °C before adding helper LCMV. Ninety hours postinfection, the passage culture was examined for luciferase expression. The packaging efficiency of each SMG was determined by dividing the relative light units (RLUs) value obtained from the passage culture 90 h postinfection by the RLUs value obtained in Figure 7a. (** *p* < 0.01, and **** *p* < 0.0001). Error bars indicate standard deviations. All experiments shown were conducted three times independently.

**Table 1 viruses-12-00896-t001:** The 5′ nucleotide sequences of untranslated regions (UTRs) of plasmids for the reverse genetics and minigenome systems.

Plasmid	5′ UTR Nucleotide Sequence (from 5′ to 3′, vRNA Polarity)
pRF-WE-SRG or SMG	CGCACCGGGGATCCTAGGCTTTTTGGATTGCGCTTTCCTTTAGGACAACTGGGTGCTGGATTCTATCCAGTAAAAGG
pRF-WE-SRG (or SMG)-5UTRΔ20–40	CGCACCGGGGATCCTAGGC---------------------TAGGACAACTGGGTGCTGGATTCTATCCAGTAAAAGG
pRF-WE-SRG (or SMG)-5UTRΔ41–60	CGCACCGGGGATCCTAGGCTTTTTGGATTGCGCTTTCCTT--------------------TTCTATCCAGTAAAAGG
pRF-WE-SRG (or SMG) -5UTRΔ60–77	CGCACCGGGGATCCTAGGCTTTTTGGATTGCGCTTTCCTTTAGGACAACTGGGTGCTGG------------------
pRF-WE-SRG (or SMG)-3UTRΔ20–38	CGCACCGGGGATCCTAGGCTTTTTGGATTGCGCTTTCCTTTAGGACAACTGGGTGCTGGATTCTATCCAGTAAAAGG
pRF-WE-SRG (or SMG)-3UTRΔ39–60	CGCACCGGGGATCCTAGGCTTTTTGGATTGCGCTTTCCTTTAGGACAACTGGGTGCTGGATTCTATCCAGTAAAAGG
pRF-WE-SRG (or SMG)-UTR-comple	CGCACCGGGGATCCTAGGCCCAAAAAATTGCGCTTTCCTTTAGGACAACTGGGTGCTGGATTCTATCCAGTAAAAGG
pRF-WE-SRG-UTR 5-3 change	CGCACCGGGGATCCTAGGCTAAA--CTAACGCGAAAATAATAGGACAACTGGGTGCTGGATTCTATCCAGTAAAAGG
pRF-WE-SRG-Δ26–40	CGCACCGGGGATCCTAGGCTTTTTG---------------TAGGACAACTGGGTGCTGGATTCTATCCAGTAAAAGG
pRF-WE-SRG-Δ20–25	CGCACCGGGGATCCTAGGC------GATTGCGCTTTCCTTTAGGACAACTGGGTGCTGGATTCTATCCAGTAAAAGG
pRF-WE-SRG-Δ20–30	CGCACCGGGGATCCTAGGC-----------CGCTTTCCTTTAGGACAACTGGGTGCTGGATTCTATCCAGTAAAAGG
pRF-WE-SRG-Δ31–40	CGCACCGGGGATCCTAGGCTTTTTGGATTG----------TAGGACAACTGGGTGCTGGATTCTATCCAGTAAAAGG

Underscores indicate mutated nucleotide sequences. Hyphens indicate deletion of nucleotide sequences in these positions.

**Table 2 viruses-12-00896-t002:** The 3′ nucleotide sequences of UTRs of plasmids for the reverse genetics and minigenome systems.

Plasmid	3′ UTR Nucleotide Sequence (from 3′ to 5′, cRNA Polarity)
pRF-WE-SRG or SMG	GCGTGTCACCTAGGATCCGTAAA--CTAACGCGAAAATAAACCTTTAAGTAACACACTGTTT
pRF-WE-SRG (or SMG)-5UTRΔ20–40	GCGTGTCACCTAGGATCCGTAAA--CTAACGCGAAAATAAACCTTTAAGTAACACACTGTTT
pRF-WE-SRG (or SMG)-5UTRΔ41–60	GCGTGTCACCTAGGATCCGTAAA--CTAACGCGAAAATAAACCTTTAAGTAACACACTGTTT
pRF-WE-SRG (or SMG)-5UTRΔ60–77	GCGTGTCACCTAGGATCCGTAAA--CTAACGCGAAAATAAACCTTTAAGTAACACACTGTTT
pRF-WE-SRG (or SMG)-3UTRΔ20–38	GCGTGTCACCTAGGATCCG---------------------ACCTTTAAGTAACACACTGTTT
pRF-WE-SRG (or SMG)-3UTRΔ39–60	GCGTGTCACCTAGGATCCGTAAA--CTAACGCGAAAATAA----------------------
pRF-WE-SRG (or SMG)-UTR-comple	GCGTGTCACCTAGGATCCG-GTTTA-TAACGCGAAAATAAACCTTTAAGTAACACACTGTTT
pRF-WE-SRG-UTR 5-3 change	GCGTGTCACCTAGGATCCGTTTTTGGATTGCGCTTTCCTTACCTTTAAGTAACACACTGTTT
pRF-WE-SRG-Δ26–40	GCGTGTCACCTAGGATCCGTAAA-----------------ACCTTTAAGTAACACACTGTTT
pRF-WE-SRG-Δ20–25	GCGTGTCACCTAGGATCCG------CTAACGCGAAAATAAACCTTTAAGTAACACACTGTTT
pRF-WE-SRG-Δ20–30	GCGTGTCACCTAGGATCCG-----------GCGAAAATAAACCTTTAAGTAACACACTGTTT
pRF-WE-SRG-Δ31–40	GCGTGTCACCTAGGATCCGTAAA-CTAAC-----------ACCTTTAAGTAACACACTGTTT

Underscores indicate mutated nucleotide sequences. Hyphens indicate deletion of nucleotide sequences in these positions.

**Table 3 viruses-12-00896-t003:** RNA secondary structure prediction of the LCMV S segment RNA produced from each plasmid.

Plasmid Name	Number of Nucleotides Forming a Panhandle Structure at the Termini		Base Pairs with High Base-Pairing Probability, Except for the 19 Base Pairs at the Termini	
	At the 5′ Terminus	At the 3′ Terminus	At the 5′ Terminus	At the 3′ Terminus
pRF-WE-SRG	45 nt	42 nt	28–33	26–31
pRF-WE-SRG-5UTRΔ20–40	19 nt	19 nt	none	none
pRF-WE-SRG-5UTRΔ41–60	36 nt	34 nt	28–33	26–31
pRF-WE-SRG-5UTRΔ60–77	45 nt	42 nt	28–33	26–31
pRF-WE-SRG-3UTRΔ20–38	27 nt	23 nt	none	none
pRF-WE-SRG-3UTRΔ39–60	36 nt	34 nt	28–33	26–31
pRF-WE-SRG-UTR-comple	45 nt	42 nt	28–33	26–31
pRF-WE-SRG-UTR 5-3 change	34 nt	36 nt	26–31	28–33
pRF-WE-SRG-Δ26–40	25 nt	25 nt	none	none
pRF-WE-SRG-Δ20–25	39 nt	38 nt	20–27	20–27
pRF-WE-SRG-Δ20–30	34 nt	33 nt	20–22	20–22
pRF-WE-SRG-Δ31–40	35 nt	32 nt	none	none

**Table 4 viruses-12-00896-t004:** Viral growth efficiency in vitro and pathogenicity in vivo for recombinant LCMVs generated from each plasmid.

Plasmid Name	Virus Name	Viral Growth Efficiency In Vitro Compared to That of rtLCMV in Vero Cell	Viral Pathogenicity In Vivo (Mortality Rate)		
			CBA/NSlc Mice	DBA/1JJmsSlc Mice	Immunity Acquired
**pRF-WE-SRG**	**rwtLCMV ^#^**		80%	60%	Acquired
pRF-WE-SRG-5UTRΔ20–40	rLCMV-5UTRΔ20–40	None	Not done (ND)	ND	ND
pRF-WE-SRG-5UTRΔ41–60	rLCMV-5UTRΔ41–60	Equal	40%	0%	Acquired
pRF-WE-SRG-5UTRΔ60–77	rLCMV-5UTRΔ60–77	Equal	80%	0%	Acquired
pRF-WE-SRG-3UTRΔ20–38	rLCMV-3UTRΔ20–38	None	ND	ND	ND
pRF-WE-SRG-3UTRΔ39–60	rLCMV-3UTRΔ39–60 ^##^	Less	0%	20%	Acquired
pRF-WE-SRG-UTR-comple	rLCMV-UTR-comple	Less	0%	0%	Acquired *
pRF-WE-SRG-UTR 5-3 change	rLCMV-UTR 5-3 change	Less	0%	0%	Acquired **
pRF-WE-SRG-Δ26–40	rLCMV-Δ26–40	Less	0%	0%	Acquired ***
pRF-WE-SRG-Δ20–25	rLCMV-Δ20–25	None	ND	ND	ND
pRF-WE-SRG-Δ20–30	rLCMV-Δ20–30	None	ND	ND	ND
pRF-WE-SRG-Δ31–40	rLCMV-Δ31–40	None	ND	ND	ND

^#^ Viral growth efficiency of rwtLCMV in Vero cells is equal that of wtLCMV. ^##^ Viral growth efficiency of rLCMV-3UTRΔ39–60 in Vero cells is equal that of wtLCMV. * One DBA/1JJmsSlc mouse infected with rLCMV-UTR-comple showed ruffled fur and weight loss. ** All DBA/1JJmsSlc mice showed ruffled fur and weight loss. *** All DBA/1JJmsSlc mice showed ruffled fur and weight loss, and one of five mice died.

**Table 5 viruses-12-00896-t005:** Mutations acquired through the serial passage in Vero cells.

Virus Name	Segment	Nt Position (vRNA Sense)	P0 *	P10 **	Gene Name	Amino Acid Change	Remarks
rLCMV-5UTRΔ60–77	L	929	C	T and C ***	L	V2091I	
rLCMV-UTR-comple	S	20	C	C and T ***	5’ UTR		The authentic nucleic acid sequence of wtLCMV at the 20th position was T.
	S	21	C	C and T ***	5’ UTR		The authentic nucleic acid sequence of wtLCMV at the 21st position was T.
	S	3355	T	T and A ***	3’ UTR		The authentic nucleic acid sequence of wtLCMV at the 3355th position was A.
	S	3356	G	G and A ***	3’ UTR		The authentic nucleic acid sequence of wtLCMV at the 3356th position was A.
rLCMV-UTR 5-3 change	S	32	A	C and A ***	5’ UTR		
	S	969	T	T and C ***	GPC	C298C	Synonymous change.
	S	3334	C	T and C ***	3’ UTR		
	L	2463	A	G and A ***	L	C1579C	Synonymous change.
	L	2524	A	G and A ***	L	V1559A	
	L	2614	A	G and A ***	L	L1529P	
	L	2689	A	G and A ***	L	F1504S	
	L	3757	C	T and C ***	L	S1148N	
	L	6698	A	G and A ***	L	L168L	Synonymous change.
	L	6699	A	G and A ***	L	L167L	Synonymous change.
rLCMV-Δ26–40	L	2708	C	T	L	E1498K	
	L	5332	T	A	L	K623M	

* P0 indicates the nucleic acid sequence of original rLCMVs. ** P10 indicates the nucleic acid sequence of serially passaged rLCMVs. *** Sequences represented by “x” and “y” in the P10 column indicate that the nucleic acid sequence of this nt position was a mixture of “x” and “y” and the major nucleic acid was “x”.

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
