# Peer review of "Analysis of the Function of the Lymphocytic Choriomeningitis Virus S Segment Untranslated Region on Growth Capacity In Vitro and on Virulence In Vivo"

_viruses, 2020, doi:10.3390/v12080896_

Round 1
Reviewer 1 Report
The authors established a minigenome and reverse genetics system for LCMV WE strain. This virus belongs to the bunyavirales order (family of arenaviridae) which contains viruses (such as Lassa virus and Rift valley fever virus) of high priority for the development of medical countermeasures according to the WHO R&D blueprint. Although related to Lassa virus, LCMV usually causes a very different disease involving the nervous system. However, infections can cause abortions in pregnant women. The relevance of this pathogen is thus given. Despite the differences LCMV has been used as a model system for the investigation of arenavirus molecular biology and for vaccine development studies as it allows experiments under lower biosafety regulations than Lassa virus. The authors report new molecular biology tools for this specific LCMV strain while for the LCMV Armstrong strain such systems already exist. They use the new tools to investigate the untranslated regions of the arenavirus small genomic segment. The genome ends contain complementary sequences, which are important for gene expression and genome replication. The authors presume that the complementarity is needed for the genomic RNA to form a panhandle structure which is also observed in viral particles. Other possible explanations are the ambisense coding strategy which simply requires the respective (complementary) sequences on both sides of the genome. It is most likely that the genome termini (19 terminal nucleotides) are bound by the viral L protein in separate binding pockets causing this panhandle structure as seen in influenza viruses. Nevertheless, the authors generated a set of UTR mutants and quantified reporter gene expression and virus packaging. They conducted animal experiments to investigate whether the mutated viruses cause a similar disease as the wildtype virus.
This study is interesting and deals with the basic understanding of the viral promoters. It also highlights a possible deviation between viability of viruses in cell culture vs. pathogenicity in the animal model, an important fact to consider. While I acknowledge the enormous efforts the authors have made to establish the tools and conduct all the experiments with a substantial number of mutants I have several problems with this study as it is now:
Major points:
1. The manuscript is very complex because of the number of mutants. The authors could be more concise both in their information content and in their style.
a) For example: Why use both GFP and Renilla luciferase as reporters in parallel? It is not necessary to show both systems/results next to each other. GFP system could go to the supplement or be left out. Or, alternatively, explain the rationale for chosing two different reporter systems.
b) A schematic depiction of the developed systems would be helpful: MG-DNA results in production of genomic sense RNA in cells (by Pol-I) which can be directly transcribed by L (assisted by N)? Otherwise readers might wonder if a genome replication step is necessary before the reporter gene can be transcribed.
2. My second point of concern is about the controls for the minigenome systems and the presented data.
a) Depending on whether this were controls or mutants tested authors transfected different amounts of DNA, which can result in different transfection efficiencies. This has not been accounted for. Which relates to the next point…
b) No normalization of the luciferase data has been performed to account for differences in transfection efficiency, cell density and pipetting errors in the transfection step.
c) What is the signal-to-noise ratio for your established minigenome system? What would be the ratio if you use a polymerase inactive L protein mutant (same amount of DNA, same size of plasmid)?
d) How often have these assays been performed? Should be mentioned in the respective figure legends.
3. The authors should be more careful when phrasing conclusions from their data or presenting already “known” facts and make sure the respective limitations are properly communicated. Several examples:
a) p2, lines 62-63: “The infection of NHPs with LCMV-WE is considered a suitable animal model for provision of insights into Lassa fever in humans.“ Please provide a reference for this statement.
b) p2, lines 63-65: “IGR…that fold into a predictable and stable secondary structure.“ Where is the evidence that there is indeed a stable secondary structure formed which has the exact form as in such a prediction? I would prefer a more cautious statement as there is no structural evidence for this secondary structure and its stability within cells (and with N protein present).
c) p13, line 372: “These mutations accumulated in S segment UTRs or L protein ORF.“ There is also one mutation in GPC. Viral replication is very balanced, not much L protein is usually expressed inside cells, otherwise toxic effects may arise (?). The observed mutations upon passaging might not display direct interaction/recognition sites inside L but enzymatic optimization complementing the impairment caused by UTR mutations – to restore a certain (non-cytotoxic) level of efficiency.
d) p19, line 523: “Furthermore, infection of NHPs with rLCMVs has provided understanding of the pathogenic mechanisms of viral hemorrhagic fever caused by arenaviruses in detail“ LCMV symptoms and pathology are very different from other arenaviruses and it is questionable how good this “model virus system” reflects on other arenaviruses such as Lassa virus. Authors should either provide a reference or phrase this sentence more carefully.
e) p19, line 520: “Our LCMV-WE reverse genetics system enabled us to clarify these characteristic differences from the virologic perspective.“ Authors should make clear that this is a future perspective and not topic of this study.
f) p19, line 531 : “The successful generation of rLCMV-UTR-comple, rLCMV-UTR 5–3 change, and rLCMV-Δ2_6_–40, and the failure to generate rLCMV-Δ2_0_–25, rLCMV-Δ2_0_–30, and rLCMV-Δ3_1_–40 suggested that both the conformation of the panhandle structure and the nucleotide sequences of these base pairs were important for the recognition of the L protein and thus affected viral propagation.“ This study contains no experiment where the recognition or interaction of the UTRs with the L protein has been investigated. Neither is there any structural data showing this hypothesized interaction outside of the terminal 19 nt. Authors should phrase this sentence more carefully.
g) p20, line 615: “In summary, we found that in addition to the 19 nucleotide base pairs in both termini of the S segment, the base pairs between the 20–40 nt in the 5‘terminus and the 20–38 nt in the 3’ terminus of the S segment formed panhandle structures with high base-pairing probabilities.“ I don’t believe this conclusion is justified. There is absolutely no structural evidence for a panhandle extending to this region. As explained already, even the terminal ends probably don’t form dsRNA but are bound in different pockets of the L protein. In the terminal 19 nt there are also non-complementary nt which are probably necessary for strand separation. What would be the functional relevance of this extended dsRNA region? Maybe for a certain step in virus amplification this panhandle is indeed present, I am not saying this conclusion is completely wrong, but it shouldn’t be generalized. Authors should rephrase this to a more cautious statement.
h) p20, line 618: “These regions affected viral propagation as they were heavily involved in viral genome transcription and replication in terms of their nucleotide sequence and conformation.“ What is the evidence for the nucleotide sequence being important in this region? No single exchanges have been done, no mutations retaining the secondary structure but changing the sequence. In a nutshell, what the authors observed is, if they cut parts of the UTRs away virus replication doesn’t work as good anymore. I think it is impossible to say whether the effects observed are due to the sequence changes or (predicted!) secondary structure changes or
Minor points:
1. p2, lines 63-65: “However, despite its uniqueness and importance, there have only been a few basic tools developed, such as a reverse genetics system, to analyze the virologic characteristics of LCMV-WE.“ This states that tools such as a reverse genetics system have been developed already. It is left unclear what is the benefit of the systems developed within this project.
2. p2, lines 72-75: “Recently, it was reported that replacement of the L segment IGR with the S segment IGR made the recombinant virus highly attenuated and induced protective immunity against a lethal wild-type LCMV (wtLCMV) challenge [28]“ Although this paper from 2015 has not really been published “recently” it shows that the IGR is of high importance for the control of viral gene expression. However, it is unclear whether the right combinations of S and L segment UTRs and IGR are necessary for proper gene expression and the herein presented study does not investigate this possibility.
3. Predictions of RNA secondary structure:
a) Usually several possible secondary structures are calculated by these kind of programs but the authors show only one result. What about other possibilities?
b) What about UTR-internal hairpins? Those have been described for influenza virus playing a role in recruitment of the genome segments to budding virions. For these you might have to enter the UTRs separately to the program.
c) It is presumed that the 5’ and the 3’ genome ends are bound by the L protein in distinct binding pockets. This means complementarity of the UTRs doesn’t imply that these regions indeed form dsRNA. These sequences are conserved and just need to be on both ends of the genome due to the ambisense coding strategy. Indeed it is likely that the 5’ end is bound as a hairpin secondary structure which is mostly stabilized by protein-RNA interactions rather than interactions between base pairs.
d) Authors should provide a link to an English-speaking homepage for the prediction program.
e) Letters in the figures are too small. 5’ and 3’ ends should be labelled in the figures.
4. p3: It is a little confusing that “IFA” is used for immunofluorescence assay instead of immunofocus assay, the latter term is used much more often, though, and IFA is the standard abbreviation for immunofocus assay in my opinion. But this is just my personal impression.
5. What is the rationale for using an immunofluorescence assay instead of/ in addition to an immunofocus assay?
6. p4: Why do the authors need an LCMV helper virus for their infectious VLP system? If the distinct UTR sequences would result in the induction of the innate immune system with different efficiencies, they could be preventing helper virus infection to a certain degree depending on the UTR mutant. How would this effect influence the data? The authors themselves point to that fact in p19, line 569 “These results suggest the 39–60 nt region in the 3’UTR may affect innate immunity; the low growth efficiency of rLCMV-3UTRΔ39–60 in A549 cells, in which the type I interferon pathway is intact, supports this hypothesis (Figure 3c).“
7. p5, line 230: “Discovery was determined using the two-stage linear step-up produced by Benjamini, Krieger, and Yekutieli, with Q = 5% [34].“ I don’t understand this sentence.
8. p5: The sequencing method should be described. Starting material is RNA, so simple sanger sequencing is not possible, right?
9. Figure 3: has some data points where there is no standard deviation shown (e.g. red curve in (b), orange curve in (c), several data points in (d)). What is the reason for this?
Have the titrations of each sample been performed in triplicates or at least duplicates?
10. Figure 4: Why have no titrations of virus growth in animal tissue/blood been performed? As it is now it is unclear whether all animals have indeed been infected (I know, unlikely in i.p. infection but still…).
11. p17, line 472: “Neither viral genome transcription nor replication was observed less strongly when cells were transfected with SMG-5UTRΔ20–40-GFP (or -luc).“ I would advise to rephrase this sentence. It is unnecessarily complicated and made me stumble.
12. p18, line 518: How similar are LCMV-WE and ARM on the amino acid level? How different are the UTRs between the two viruses?
13. p19, line 537: “The reason why rLCMV-Δ26–40 successfully propagated was not clarified in this study, but the conservation of base pairs comprising nt 20–25 in both termini and the RNA conformation may be advantageous to propagation.“ If this was the case then any sequence (provided the base pairing is intact) should work in this region of the UTRs, right?
14. p 19: Authors compare to TBEV, which has a completely different replication strategy (positive sense RNA virus). What about more closely related viruses (other families within bunyavirales or even articulavirales)?
15. p20, line 596: Please define “C-terminal domain of the L protein”.
16. p21: “the other LCMV S segment UTRs“ - Should be more concisely named “UTR regions” instead of UTRs.
17. Typos and abbreviations:
Virological instead of virologic (p2, line 65)
Fig 7c: lower right panel labeling, GFP instead of HFP
Multiplicity of infection should be MOI – makes it obvious that this is an abbreviation (in contrast to moi)
Reviewer 2 Report
The report from Taniguchi et al. describes studies on the properties of recombinant Lymphocytic Choriomeningitis Viruses (LCMV) and minigenome derived from it in vitro and in mice. This is clearly written work and very useful to progress understanding of virulent determinant of LCMV and for proof-of concept development of live attenuated LCMV vaccine candidates.
Consider providing (could be supplementary figure) predicted folded structures with their delta G for S segment mutants examined in this study
The cytotoxicity of minigenome should be addressed as controls in the experiments
Round 2
Reviewer 1 Report
First of all, thanks to the authors for their detailed and patient responses and explanations to all the points and questions I raised. Overall, I am satisfied with the manuscript in its current form. I only found some minor errors which are listed below. I would advise a final cautious spell check.
- PCAGGS should be pCAGGS
- fluorescent microscope - should be fluorescence microscope
- Page 6, Westernblot paragraph:
a) "Three hundred microliters of cell lysate were mixed with 100 μL 2- mercaptoethanol, ..." I would bet you meant 4xSDS sample buffer containing x mM 2-mercaptoethanol.
b) antibody dilutions of primary and secondary antibodies should be given
c) "the membrane...and then detected using..." The grammar of this sentence should be corrected.
Final remark: Titration of animal blood to prove that an animal was indeed infected should be a standard procedure for any animal experiment, also in respect of getting the most information and significant data out of animal experiments in order to justify the sacrifice. Results in figure 5 don't show that all mice were infected in experiments of figure 5. However, I agree that this is difficult to add to the study at this point.
Author Response
Response to reviewer 1’s comments
First of all, thanks to the authors for their detailed and patient responses and explanations to all the points and questions I raised. Overall, I am satisfied with the manuscript in its current form. I only found some minor errors which are listed below. I would advise a final cautious spell check.
- PCAGGS should be pCAGGS
- fluorescent microscope - should be fluorescence microscope
- Page 6, Westernblot paragraph:
- a) "Three hundred microliters of cell lysate were mixed with 100 μL 2- mercaptoethanol, ..." I would bet you meant 4xSDS sample buffer containing x mM 2-mercaptoethanol.
- b) antibody dilutions of primary and secondary antibodies should be given
- c) "the membrane...and then detected using..." The grammar of this sentence should be corrected.
- We are pleased to hear that Reviewer 1 was satisfied with the revised manuscript. Again, we really appreciate the time and effort that Reviewer 1 has dedicated to providing insightful feedback on ways to strengthen our paper. We corrected the errors you listed below and some errors we found. The corrected parts are highlighted in yellow.
Final remark: Titration of animal blood to prove that an animal was indeed infected should be a standard procedure for any animal experiment, also in respect of getting the most information and significant data out of animal experiments in order to justify the sacrifice. Results in figure 5 don't show that all mice were infected in experiments of figure 5. However, I agree that this is difficult to add to the study at this point.
- Thank you for pointing it out again. Unfortunately, we did not collect blood samples this time. We would like to take your remark into consideration to the next animal experiments.